# Development and Validation of Child Observation Checklist for Early Screening of Young Children with Special Needs

**DOI:** 10.3390/healthcare13020148

**Published:** 2025-01-14

**Authors:** Anna Na Na Hui, Angela Fung Ying Siu, Cynthia Leung, Wally Karnilowicz

**Affiliations:** 1Department of Social and Beahvioural Sciences, City University of Hong Kong, Hong Kong; 2Department of Educational Psychology, The Chinese University of Hong Kong, Hong Kong; afysiu@cuhk.edu.hk; 3Mitchell Institute, Victoria University, Melbourne, VIC 3000, Australia; cynthia.leung@vu.edu.au; 4Institute of Health and Sport, Victoria University, Melbourne, VIC 8001, Australia; waldok81@gmail.com

**Keywords:** early screening, special needs, psychometric validation, teachers, early childhood education, family support

## Abstract

**Background:** Families with young children with developmental disabilities often express concerns about delays in the identification of these and extended waiting times for obtaining assessments, learning support, and rehabilitation services. The identification process must and can be enhanced if preschool teachers have instrumental measures to detect early-stage developmental disabilities and adjustment difficulties in preschool children and, if necessary, to provide these children with prompt and effective support services. **Aims:** The aim with this study was to develop and validate a screening checklist for teachers to detect developmental delays and adjustment difficulties in Chinese preschool children in Hong Kong. **Methods:** The participants included 1085 children aged two to six years (including 365 children who were receiving rehabilitation services at the time of the study) and their preschool teachers. The teachers were requested to complete the screening checklist and the Strengths and Difficulties Questionnaire. **Results:** The results of the Rasch and Confirmatory Factor Analyses supported the unidimensionality of the checklist, with the validated version consisting of fifteen items and five factors. The checklist score was negatively correlated with children’s behavioral problems and effectively differentiated children of different ages and between children receiving and not receiving rehabilitation services. The reliability estimates (internal consistency and test–retest) of this revised checklist were above 0.70. **Conclusions:** The 15-item checklist is a promising screening instrument for the identification of developmental disabilities and adjustment problems among Chinese preschool children. The use of the checklist has accelerated access to rehabilitation services for children and family caregivers.

## 1. Introduction

The accurate and effective identification of developmental disabilities in one or multiple domains with respect to age is crucial in identifying young children with special needs. The early identification of young children with developmental disabilities helps ensure the provision of timely support and intervention [1]. The development and practice of using a valid screening tool that enables such early identification of developmental disabilities is essential in successfully accessing and using early intervention services [2,3]. A screening tool with strong psychometric properties (e.g., sensitivity/specificity, reliability, and validity) would be particularly useful in identifying children’s developmental disabilities and associated concerns [4,5]. The emotional and behavioral domains are also very important areas for achieving an understanding of a child’s atypical development, particularly because problems that develop at an early age tend to persist over time [6,7]. Given the persistent nature of such delays in child development, there is an urgent need to enable early identification and interventions in order to successfully attend to social–emotional and behavioral problems in young children, as well as to support their parents [8].

In Hong Kong, approximately 88.4% of children aged between 3 and 6 years are enrolled in preschool programs [9]. In 2021, approximately 50,000 children aged 2 years were enrolled in prenursery programs. The government, informed by the positive outcomes of a pilot scheme, regularized the On-site Preschool Rehabilitation Services (OPRS) program in 2018 to provide school-based rehabilitation services and training for children with special needs attending mainstream preschools [10]. There are other available rehabilitation services for preschool children in Hong Kong, including the Early Education and Training Center (EETC), Integrated Program (IP), and Training Subsidy Program (TSP) [11].

To be eligible for government-funded rehabilitation services, children are assessed and then recommended for such services by pediatricians and/or clinical/educational psychologists. In 2020/2021, the Pilot Project on Tier 1 Support Services (Pilot Project on Tier 1 scheme) in Kindergartens/Kindergarten-cum-Child Care Centers (KGs/KG-cum-CCCs) was made available to children under 6 years of age studying in KGs/KG-cum-CCCs who (i) are awaiting assessment by the Child Assessment Centers (CACs) of the Department of Health, Hong Kong SAR Government, (ii) have been assessed by CACs to have borderline developmental difficulties but are not eligible for subventionary preschool rehabilitation services, or (iii) have learning, social interaction, behavioral, or emotional difficulties. In 2023, the government integrated Tier 1 support services with OPRS, and based on the expertise of a multidisciplinary team, they provided comprehensive school-based services to preschool children with different levels of needs [12]. Overall, it is important that early childhood educators identify children with special needs and provide timely and effective intervention services [10,13].

### 1.1. The Need for the Development and Validation of a Culturally Relevant Measure

Instead of standardized assessments by clinicians, parent or teacher ratings of children’s performance in various developmental domains in the format of a behavioral checklist are more convenient. For instance, the Ages and Stages Questionnaire—Third edition (ASQ-3) has been adapted for young Chinese children. The ASQ-3 is a developmental screening tool for children aged 0–5 years. It is completed by parents and teachers to identify young children who are prone to developmental delays [14]. It also has strong psychometric properties [15]. However, the translated Chinese versions of the ASQ-3 that were adopted in Taiwan and China are less accurate in identifying Chinese children with special needs [16,17]. For example, there is cultural disparity in the development of fine and gross motor skills between Chinese children and children in the United States: fine motor skills tend to be more developed in Chinese children due in large part to the fine motor coordination that is required in their early use of chopsticks [18]. Developmental checklists also rely on descriptors that may be interpreted differently across cultures. Thus, screening tools that originate from one culture may not be entirely applicable for use in other cultures [19]; the same items may carry different social meanings across variable cultural contexts [20,21,22]. Of concern is the cultural fairness and validity of applying developmental screening tools that were developed in a Western culture to other social and cultural contexts [23]. Cultures have different values regarding skill development in children and convey these values differently [19].

Despite the need for culturally fair and valid early developmental screening tools for use with Chinese preschoolers, there are few locally designed developmental checklists for the early screening of young children. An exception is the Taipei City Developmental Checklist for Preschoolers (second version [Taipei II]), which was developed locally and used to assess young Chinese children in Taiwan [24]. Practitioners, parents, or teachers use the screening checklist to identify young children with potential developmental delays or disabilities. The Taipei II covers five areas of child development: cognitive, fine motor, gross motor, language and communication, and socioemotional development. However, the Taipei II has inherent weaknesses, including the use of a dichotomous scale that is restricted to determining the presence or absence of certain behaviors. Thus, the precision of the assessment is limited and does not adequately reflect children’s performance [13].

### 1.2. Hong Kong Screening Tools for Preschool Children

Two available measurements can be considered as screening tools because they are normed for preschoolers in Hong Kong. The Hong Kong Early Child Development Scale (HKECDS) is a locally developed assessment tool for use with typically developing Chinese preschoolers in Hong Kong [23]. The full version of the HKECDS comprises 190 items covering nine domains: personal, social, and self-care; language development; preacademic learning; cognitive development; gross motor; fine motor; physical fitness; moral development; society and environment. It is therefore often adopted as a summative measurement for child outcomes [23,25]. However, the time and labor costs in administering the test are significant, making it less viable as a universal screening tool for teachers’ use. The HKSPC Developmental Checklist (HKSPC-DC), developed by the Hong Kong Society for the Protection of Children (HKSPC), was developed to enable preschool teachers to detect children who are at risk of developmental delays through daily observations of children’s functioning. This assessment comprises four developmental aspects: cognitive, language (expressive and receptive language), motor (fine and gross motor), and self-care and social skills. There are four sets of scales, one of which corresponds to each preschool level (N to K3) and determines children’s grade-based achievements. Each set contains 61 to 83 items. The HKSPC-DC facilitates the early identification of Chinese preschool children who have developmental deficits. One of its strengths is that it is a comprehensive developmental checklist covering Hong Kong Chinese children’s developmental needs more specifically than checklists that are developed in Western cultures do [13]. However, its factor structure has not been confirmed, and its measurement properties are unclear.

However, both of the above scales may be considered overly long and time-consuming. As proposed by Sharma [26], the length of a questionnaire is important, as it impacts the respondents’ attention and interest. The longer the survey is, the more likely it is that the quality, reliability, and response rates could be affected. With reference to the 27-item Primary One Screening Checklist that was previously developed by Leung and colleagues [27], it may be more user-friendly and time-effective if a checklist can be completed by teachers within 10 to 15 min and includes items from various developmental domains.

The Child Observation Checklist (COC) was developed by a group of educational psychologists from nongovernmental organizations (NGOs) taking part in the Pilot Tier 1 Scheme. It was constructed for use by preschool teachers and OPRS professionals in Hong Kong. The project was initiated by the Hong Kong government to equip teachers and childcare workers with the required knowledge and skills to conduct an initial screening of children with mild developmental problems. The screening checklist is intended for and best used by teachers, given their professional training, strong reading skills, and numerous opportunities to observe children’s interactions with their peers [28]. This approach directly and positively impacts the accuracy of referrals.

### 1.3. The Present Study

This study aimed to validate the COC for use in the early identification of preschool children with mild or transient developmental or adjustment difficulties. The checklist assessment provides information for use by teachers of Chinese preschool children and allied health professionals in their referral of children for preschool rehabilitation services. This allows teachers and professionals to quickly detect problematic issues and provide timely support for facilitating child development.

The framework of the checklist was developed through discussions among professionals, predominantly educational psychologists from the six NGOs who participated in the Pilot Tier 1 scheme. The development of the framework leveraged expertise from two clinical psychologists, three preschool teachers, two pediatricians, two occupational therapists, and two physiotherapists from local government departments, i.e., the Social Welfare Department and Child Assessment Service. This group agreed that the framework should focus on children’s adjustment difficulties that are observable and associated with the child’s various special needs. Local and foreign assessment tools on adjustment or school readiness for young children were examined. The finalized COC consisted of 76 items within 5 subscales, including 13 items for learning adaptation (LA), e.g., routine, switch, focus, and engagement; 20 items for social adaptation (SA), e.g., joint focus, communication, and interaction; 13 items for behavioral and emotional regulation (BER), e.g., self-control, emotional relief, and problem solving; 12 items for daily self-care (SC); and 18 items for gross and fine motor performance (MP). Each item was rated using a five-point Likert scale ranging from 0 (never, meaning that it did not happen) to 4 (always, meaning that it happened five times or more per week). The rating assessed the child’s performance in the past four weeks across each of the five subscales.

To test the validity of this new screening tool, the hypotheses were as follows:The COC score is positively correlated with prosocial behavior and negatively correlated with behavioral problems.Children in higher grades (older children) receive higher scores than those in lower grades (younger children).Children not receiving rehabilitation services achieve higher scores than those receiving rehabilitation services (OPRS or Tier 1 services).

## 2. Materials and Methods

### 2.1. Participants

The participants included a random sample of 1085 children who were drawn from 67 preschools participating in the Pilot Tier 1 scheme as of September 2022. The participating preschools also received OPRS. The number of participating children from each preschool ranged from 3 to 35, with an average of 16.2 children per preschool. The mean age of the children was 4.03 years (SD = 1.09, range: 2 to 7). Their teachers were invited to complete the COC and Strengths and Difficulties Questionnaire (SDQ). The distribution of the children’s characteristics is shown in Table 1.

Amongst the participants, 720 (66.4%) children did not receive support services; 180 (16.6%) received Tier 1 services; and 185 (17.1%) received subventionary preschool rehabilitation services, i.e., OPRS, IP, EETC, TSP, or other services (see Table 1). There was a greater percentage (25.3%) of K3 children than nursery children (2.2%) receiving OPRS/IP/EETC/TSP/other services (χ^2^ (6) = 34.87, *p* < 0.001).

### 2.2. Measures

Teachers were invited to complete the COC and SDQ questionnaires for the children in their classrooms. The COC, previously described, was completed by teachers approximately two months after the commencement of the first school term (approximately in December).

The SDQ is a brief behavioral screening questionnaire for children and adolescents aged four to sixteen years. It consists of five subscales: emotional symptoms; behavioral problems; hyperactivity/inattention; peer relationship problems; and prosocial behavior [29,30]. Each item is rated on a 3-point rating scale from 1 (not true) to 3 (definitely true), with higher scores indicating a greater endorsement of the behavioral domain. The total problematic behavior score was computed by summing the raw scores from the emotional symptoms, behavioral problems, hyperactivity/inattention, and peer relationship problems subscales. The Chinese version of the scale was validated for children aged 6 to 12 years. In the validation study, the reliability estimates for the total problematic behavior scale and all subscales (except for peer relationship problems) were above 0.70. The reliability of the peer relationship problems subscale was 0.45 [30].

### 2.3. Procedures

This study was approved by the Human and Artefacts Ethics Sub-Committee of the affiliated university of the first author. Three boys and three girls from each grade were randomly selected in each participating preschool. An invitation letter with a parental consent form was then sent to 1628 parents through the preschools. Written consent was obtained from parents during November 2022. Data were collected from November 2022 to March 2023. The parents’ children represented 67 of the 87 preschools that participated in the Tier 1 and OPRS schemes. The class teachers of these children also consented to participate in the study. A total of 1085 teacher responses (from 67 schools) were received from February to March 2023. A measure of the test–retest reliability was obtained through a repeat completion of the COC in March 2023 by the class teachers of 193 randomly selected students.

### 2.4. Data Analysis

Several methods were used to examine the validity of the COC [31]. A Confirmatory Factor Analysis (CFA) determined the construct validity of the COC; several assessment measures have been translated and validated in Chinese samples using CFA [32]. The convergent validity involves measuring a construct’s correlation with related measures [33]. This was examined in terms of the correlation (Pearson) between the COC score and children’s behaviors. Local research shows that school readiness and learning motivation correlate positively with prosocial behavior and negatively with problematic behavior, as measured by means of the SDQ [3,34]. The construct validity of the checklist was tested through the known-group validity method [35], where we administered the COC to two or more groups that were known to obtain different scores for a construct, as measured by means of the COC. In this case, we conducted an assessment of the COC’s ability to differentiate children from different age groups and children receiving rehabilitation services from those who did not. It is expected that younger children will achieve lower scores than older children, and children receiving rehabilitation services will achieve lower scores than those who are not receiving such services.

The measurement properties of the COC were examined using a Rasch analysis. As the items are summed to form a total score, unidimensionality was examined using infit and outfit statistics, point measure correlations, and principal component analysis (PCA) of the residuals that remained after the extraction of the linear Rasch measure. A Wright map was used to examine the targeting of the checklist to determine whether the difficulty level of the items targeted the ability of the children. The category functioning of the Likert scale was also examined to determine whether teachers could differentiate between the scale points. Modifications were made where appropriate based on theory-driven considerations of the analysis results.

Reliability was assessed through measures of internal consistency (Cronbach’s alpha) and test–retest reliability (intraclass correlation). Reliability estimates above 0.70 are regarded as moderate or fair, and estimates above 0.80 are regarded as good [33]. It is generally agreed that a higher test–retest correlation indicates better instrument reliability [36].

As a screening checklist, validity was also examined in terms of the sensitivity and specificity of the COC in identifying children who may need support services. The sensitivity and specificity were examined through the receiver operating characteristic (ROC) curve method [37]. The sensitivity + specificity should be at least 1.5 for a test to be considered useful [38]. The positive likelihood value (LR+) is a good indicator for ruling-in a diagnosis, with higher values being stronger indicators. The negative likelihood value (LR−) is a good indicator for ruling out a diagnosis, with lower values being stronger indicators [39]. The odds ratio between LR+ and LR− is a measure of the accuracy of prediction, and higher values indicate better predictions [13]. In terms of diagnostic accuracy, an area under the ROC curve (AUC) of 0.6 to 0.7 is regarded as sufficient, a value of 0.7 to 0.8 is regarded as good, and a value of 0.8 to 0.9 is regarded as very good [39]. Youden’s index is a measure of screening accuracy, and higher values indicate better accuracy [13].

A Rasch analysis using Winsteps 5.4.2.0 was used to examine the measurement properties of the COC, including unidimensionality, targeting, and category functioning [40]. The CFA was conducted using AMOS 29. Differences between grade/age levels and child service status (receiving rehabilitation services or not) were analyzed using independent *t*-tests and Analysis of Variance (ANOVA). These procedures, together with the ROC and estimates of reliability, were analyzed using SPSS 29.

## 3. Results

### 3.1. The 5-Factor, 76-Item Version (COC)

#### 3.1.1. Confirmatory Factor Analysis

The validity associated with each of the five constructs was assessed using CFA. The initial analysis tested the validity of the 5-factor, 76-item model. Preliminary tests of kurtosis (3.79 to −1.28) and skewness (−0.30 to −2.14) were largely satisfactory and justified the use of maximum likelihood estimation with the CFA [40]. The fit statistics associated with the 5-factor, 76-item model were poor. For example, the Chi Square (χ^2^) (2, 764) = 24,588.710, *p* < 0.001. In the Root Mean Squared Error of Approximation addition, the Comparative Fit Index (CFI) = 0.771 and the Tucker–Lewis Index (TLI) = 0.764, far less than the recommended cutoff values of greater than 0.950 [41], and the RMSEA of 0.085 was far greater than the ‘good’ cutoff score of <0.06 [42].

#### 3.1.2. Rasch Analysis

The unidimensionality was evaluated through an examination of the infit and outfit mean square statistics, point measure correlations, and PCA. For the infit and outfit mean square statistics, using a cutoff of 0.6 to 1.4, there were 11 items with infit statistics that were outside the recommended range (SA10, BER1, BER2, BER3, SC5, SC6, SC8, SC9, MP3, MP8, and MP17) and 9 items with outfit statistics that were outside the recommended range (LA11, SA10, BER1, BER2, BER3, SC6, SC8, MP8, and MP17) [40]. All point measure correlations were positive. For the PCA results, the criteria for unidimensionality were as follows: (i) the variance explained by measures of 40% or more; (ii) the variance explained by the first principal component of the residuals, which must be 15% or less; and (iii) the ratio of the variance in measures to the variance in the first principal component of the residuals, which must be 3:1 or more [43]. In the present case, the variance explained by the measures was 58.0%, and the variance explained by the first principal component of the residuals was 4.5%. The ratio of the variance in measures to the variance in the first principal component of the residuals was 12.89:1, fulfilling the criteria set out [43]. The mean infit mean square was 1.06 (sd: −0.23), and the mean outfit fit mean square was 1.05 (sd: −0.25). The person reliability was 0.97, and the person separation was 6.03. The item reliability was 0.99, and the item separation was 12.92. The item map indicated that the items were concentrated on the lower end and that there were not enough items targeting children with higher abilities (see Figure 1).

#### 3.1.3. Reliability

The internal consistency of the COC was examined using Cronbach’s alpha. All estimates were above 0.90. The test–retest reliability was estimated using the intraclass correlation coefficient (ICC, two-way random, consistency), and all estimates were above 0.80, with *p* < 0.001 in all cases. The details are presented in Table 2.

### 3.2. The 5-Factor, 15-Item Version (COC-15)

#### 3.2.1. Confirmatory Factor Analysis

The 76-item, 5-factor model was revised because of the unsatisfactory outcomes of the CFA and Rasch analyses. Based on the results of a further Rasch analysis, 13 items with unsatisfactory infit and outfit statistics were removed from the original 76-item model, with the resulting 63-item version having fit statistics within 0.6 to 1.4. The construct validity of the new 63-item model was further examined using CFA. Guided by the process of modification indices and the associated examination of factor loadings and covariances, the final model version comprised 15 items (COC-15 as seen in Appendix A) and was acceptable in terms of model fit (see Figure 2). The fit statistics, which included the χ^2^ value, the CFI, the TLI, and the RMSEA associated with the 5-factor, 15-item model (COC-15), supported the fit of the model. The χ^2^ of 330.90 (79) with *p* < 0.001; the CFI of 0.098; and the TLI of 0.98 were all greater than the recommended cutoff value of 0.90 [42]. The RMSEA of 0.05 also supported a ‘good’ model fit, defined as being below the accepted cutoff of <0.06 [44].

#### 3.2.2. Rasch Analysis

The unidimensionality was evaluated through an examination of the infit and outfit mean square statistics, point measure correlations, and PCA. For the infit and outfit mean square statistics, all items were within 0.6 to 1.4 [40]. All point measure correlations were positive. The PCA results revealed that the variance explained by the measures was 62.50%, and the variance explained by the first principal component of the residuals was 7.1%. The ratio of the variance in measures to the variance in the first principal component of the residuals was 8.81:1, fulfilling the criteria described earlier [44].

For category functioning, although the average measures increased from −1.59 for category 0 to 2.93 for category 4, and there were more than 10 responses for each category, the threshold calibrations were less than 1.4 logits apart. This suggested that the teachers might not have been able to adequately distinguish between the five categories. The mean infit mean square was 1.00 (sd: −0.17) and the mean outfit mean square was 1.01 (sd: −0.14). The person reliability was 0.92 and the person separation was 3.47. The item reliability was 0.99 and the item separation was 11.55. The item map showed that the items were concentrated on the lower end, and there were not enough items targeting students with higher abilities (see Figure 3). It was justified to have items on the lower end to ensure that the tool was appropriately tailored to the ability levels of the target students and was effective in identifying students who required additional support.

#### 3.2.3. Reliability

The internal consistency of the COC-15 was examined using Cronbach’s alpha, with estimates that were all above 0.70; the test–retest reliability was estimated using the ICC (two-way random, consistency), and all estimates were above 0.80, with *p* < 0.001 in all cases (see Table 2).

### 3.3. Validity

#### 3.3.1. Association with Student Behavior

The total COC-15 and subscale scores correlated positively with SDQ prosocial behavior (*p* < 0.001) and negatively with SDQ behavioral problems (*p* < 0.001); the same pattern was observed in the total sample and based on grade level (see Table 3).

#### 3.3.2. Differentiation of Students Based on Service Status

The children were categorized into three groups as described in the participant section: (1) children receiving preschool rehabilitation services such as OPRS, IP, EETC, TSP, and other services on an individual level (Tier 2); (2) children receiving Tier 1 services on a classroom level (Tier 1); and (3) children receiving no service (no-service). The ANOVA results indicated that there was a significant difference in COC-15 scores based on the service status within the complete sample, with F (2, 1082) = 68.29 and *p* < 0.001; for the K1 students, with F (2, 342) = 36.88 and *p* < 0.001; for the K2 students, with F (2, 301) = 58.70 and *p* < 0.001; and for the K3 students, with F (2, 341) = 53.12 and *p* < 0.001. However, among the nursery students, F (2, 89) = 2.96 and *p* = 0.057. The post hoc test (Scheffe) results indicated that the COC-15 differentiated Tier 1 and Tier 2 students from the no-service group among K1, K2, and K3 students (see Table 4). Overall, the no-service group students were rated higher by their teachers than the Tier 1 and Tier 2 students were.

An independent *t*-test was used to examine the ability of the COC-15 to differentiate between the no-service group and the combined Tier 1/Tier 2 group. The results were significant for the complete sample, with t (1083) = 11.64 and *p* < 0.001; the nursery group, with t (90) = 2.34 and *p* = 0.021; the K1 group, with t (343) = 8.29 and *p* < 0.001; the K2 group, with t (302) = 10.85 and *p* < 0.001; and the K3 group, with t (342) = 10.31 and *p* < 0.001. In all cases, teachers rated the no-service group higher than the Tier 1/Tier 2 group (see Table 4).

#### 3.3.3. Differentiation of Students Based on Grade Level (Age)

The ANOVA results were significant for differentiation based on grade level for LA, with F (3, 1081) = 25.42 and *p* < 0.001; for SA, with F (3, 1081) = 53.08 and *p* < 0.001; for BER, with F (3, 1081) = 83.11 and *p* < 0.001; for SC, with F (3, 1081) = 203.15 and *p* < 0.001; for MP, with F (3, 1081) = 135.07 and *p* < 0.001; and for the total score, with F (3, 1081) = 112.97 and *p* < 0.001. The post hoc analyses indicated that there were significant differences across grades, with higher ratings for students in higher grades (see Table 5).

#### 3.3.4. Sensitivity and Specificity

An ROC analysis was performed separately for each grade to identify the cutoff point that could best identify students requiring support services in each grade. The teacher’s report of the children’s service status (no service vs. Tier 1/Tier 2) was the state variable (see Table 6).

## 4. Discussion

This study aimed to develop and validate an observation measure for the early identification of mild or transient developmental or adjustment difficulties among preschool children. The assessments are for use by kindergarten teachers of Chinese preschool children and allied health professionals in determining access to preschool rehabilitation services. Since the model fit for the original 76-item version was poor, a final model version comprising 15 items (COC-15) was adopted. The reliability of the COC-15 was found to be satisfactory, as was the test–retest reliability. The Rasch analysis results indicated that the COC-15 conforms to the requirements of unidimensionality, justifying the summation of item scores to form a total score [44].

Hypothesis one on the correlation between COC-15 score and child behavior was supported. The COC-15 score was positively correlated with the SDQ prosocial behavior score and negatively correlated with the SDQ total problematic behavior score. These results provide support for convergent validity. The moderately strong and positive correlations of all the subscales of COC-15 and the SDQ prosocial subscale are comparable to the moderately strong correlations of teacher-rated positive and negative behaviors of children aged between 3 and 6 based on the SDQ prosocial subscale [45]. The moderately strong but negative correlations of all the COC-15 subscales and the SDQ problematic behavior subscale are also consistent with findings from other validation studies [46,47,48].

Hypothesis two on age differentiation was supported. The COC was able to differentiate children from different age groups, with those in higher grades receiving higher scores on the COC-15. The effect sizes of the mean differences in COC-15 total scores among different grades are largest between the nursery group and the K3 group. Age differentiation has also commonly been found in other validation studies of screening checklists [13,46]. As children grow older, their abilities in language and SA, BER, self-care, and motor skills also improve. The latter two areas exhibit the most observable differences in relation to each grade level.

Hypothesis three on the differentiation of children based on service status was supported. The COC-15 differentiated children who were receiving rehabilitation services from children who were not receiving rehabilitation services. Although there are not enough items to measure the performance of children with high levels of ability, the COC-15 focuses on screening for the eligibility of children with developmental disabilities for rehabilitation services, i.e., with lower ability ranges. These results provided support for criterion-related validity. The COC-15 is useful for providing teachers with more information to inform their decisions when identifying children with developmental disabilities and/or determining their need for early intervention programs. The length of the COC-15 is adequate according to Sharma [26]. This 15-item tool is able to maintain the attention and interest of respondents, meaning that a good response rate and reliability can be ensured.

Although the diagnostic accuracy was acceptable for K2 and K3 children (i.e., children aged 4 and 5, respectively), it was less ideal for K1 children (i.e., those who are aged 3). For K1 children, given that the COC-15 was administered approximately two to three months after preschool entry, the children may need time to adjust to the preschool environment, and further monitoring of the progress of the children would be necessary. The sample size of the nursery group was small, and the results need to be interpreted with caution.

The COC-15 is psychometrically sound and culturally relevant. The items were generated through expert discussion and extensive literature reviews and by adapting items from existing measures. Items were also selected based on their validity and assessed using CFA. Most importantly, the COC-15 is helpful in differentiating preschool children who are not in need of rehabilitation services from those who are receiving services and detecting variations between students in different grades. The COC-15 is particularly useful for identifying children who might need to be involved in early intervention programs. In addition, the COC-15 consists of only 15 items and is not too demanding of teacher time. It is therefore more practicable and acceptable for use in schools.

Our study is not without limitations. First, performance-based and observational data are crucial for validating these measures [49,50]. Due to time limitations and associated constraints, it was not possible to provide individual assessments for each child for the purpose of triangulation. Second, the number of kindergartens that were included in the study was limited to approximately one-tenth of the total number of kindergartens in Hong Kong; a larger representative sample is required and would be useful in establishing a local norm. Third, the COC-15 items are loaded more on the lower end of the children’s ability spectrum. Although this approach is not inappropriate for the present case of using the COC-15 to identify children with developmental disabilities for eligibility to receive an intervention, it may result in a ceiling effect when used in other studies to examine factors relating to child development among children with a range of abilities.

The COC-15 is convenient for teachers to use. The items may also provide useful information for caregivers about the early signs of developmental disabilities. This screening tool is currently being used by the school-based rehabilitation service teams and has effectively shortened the waiting time for receiving learning support and rehabilitation services. In-class learning support and pedagogical accommodation can be provided to children who have been waiting for a standardized assessment to be conducted by the Child Assessment Service at the Department of Health. Family caregivers are also provided with consultation and counseling sessions to enhance their understanding of special education needs and the holistic development of young children, facilitate their parenting skills and parent–child relationships, and strengthen their family support, as well alleviate parental stress [51].

## 5. Conclusions

The COC-15 is a potentially valuable instrument for screening preschool children with developmental disabilities and determining their eligibility for intervention services. It has demonstrated sound validity through CFA, good correlation with child behavior, and an ability to differentiate children from different age groups and children who are receiving rehabilitation services from those who are not. In terms of its measurement properties, unidimensionality is supported, and the reliability estimates are above 0.70 in all cases. There is also support for its diagnostic accuracy. Importantly, the COC-15 serves as a useful tool, enabling professionals to understand the needs of children and decide on services that are adjusted to meet the preschool child’s needs. However, decisions should not be made solely based on the COC-15 scores. Decisions should also be made considering the context of the child, along with observation and discussions with teachers and parents and the use of other assessment tools as needed. The child’s progress should be observed and monitored continuously to provide the best possible and most beneficial service for the child.

## Figures and Tables

**Figure 1 healthcare-13-00148-f001:**
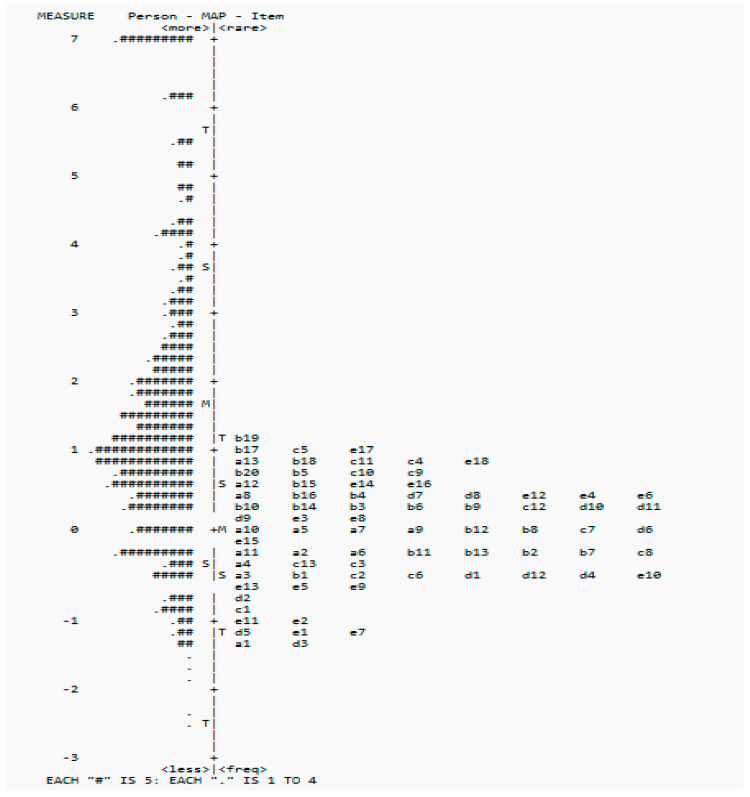
**Wright Map of Person (#) and Items** of the COC. Note: Each “#” is 5.

**Figure 2 healthcare-13-00148-f002:**
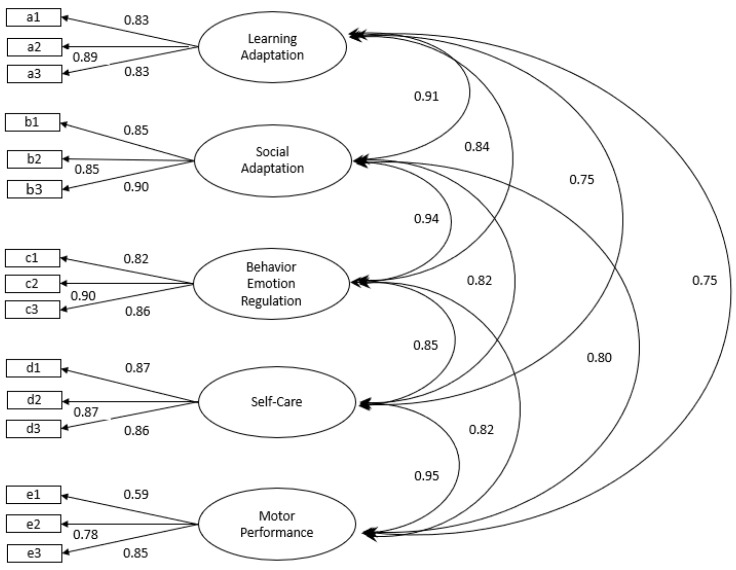
CFA 5-factor, 15-item model.

**Figure 3 healthcare-13-00148-f003:**
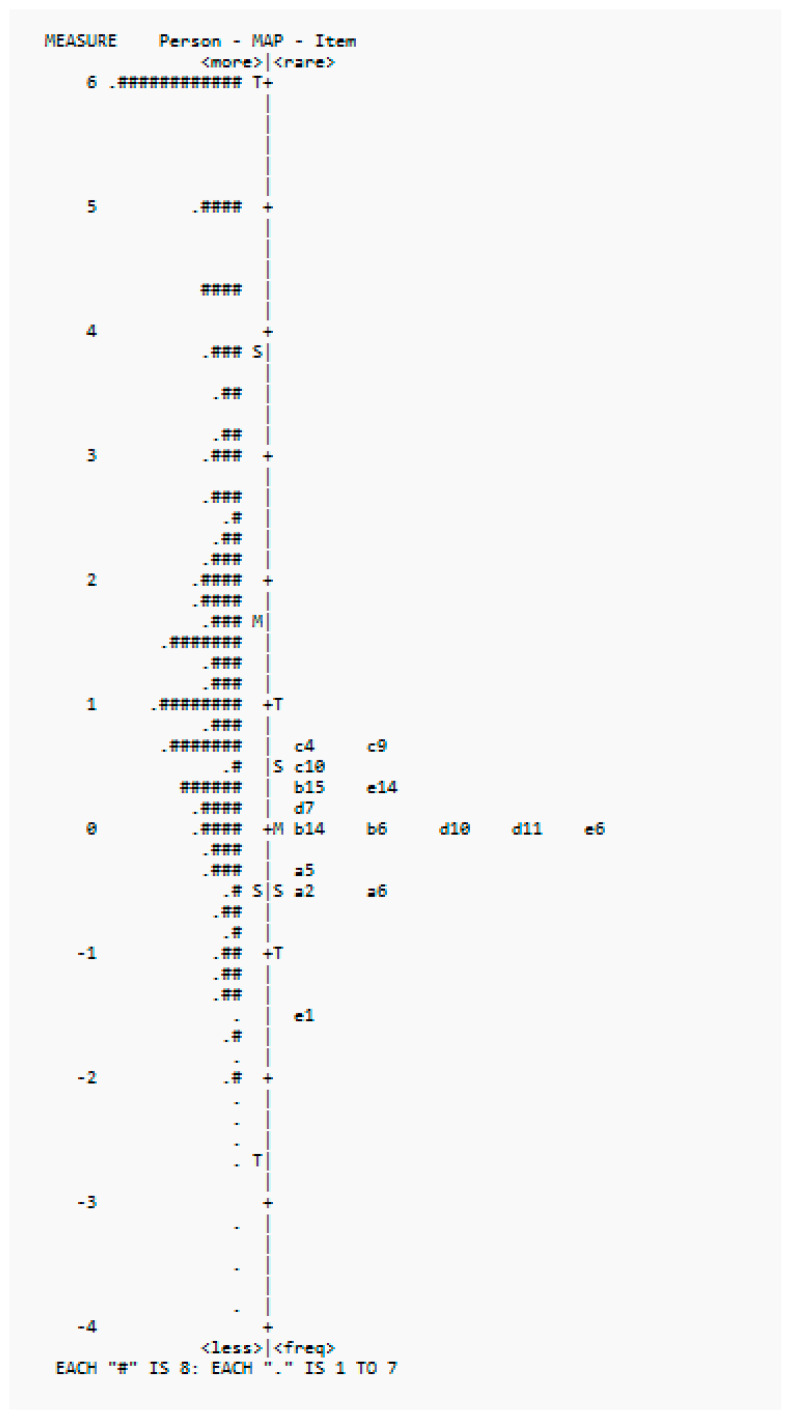
**Wright Map of Person (#) and Items** of the COC-15. Note: Each “#” is 8.

**Table 1 healthcare-13-00148-t001:** Distribution of children’s characteristics: sex, grade, and service status.

	Sex		Service Status			Total
Grade	Boys	Girls	Not receiving services	Tier 1	Tier 2	Total
Nursery	47	45	70	20	2	92
K1	185	160	242	57	46	345
K2	152	152	203	51	50	304
K3	186	158	205	52	87	344
Total	570	515	720	180	185	1085

Note: Tier 2 includes children receiving services from the On-site Preschool Rehabilitation Services, Integrated Program, Early Education and Training Center, Training Subsidy Program, or other providers.

**Table 2 healthcare-13-00148-t002:** Reliability estimates of the COC and COC-15.

	Internal Consistency (*n* = 1085) Cronbach’s Alpha		Test–Retest Reliability (*n* = 147) Intraclass Correlation Coefficient			
	COC	COC-15	COC	95% CI	COC-15	95% CI
LA	0.96	0.88	0.91	[0.87, 0.93]	0.87	[0.82, 0.90]
SA	0.98	0.90	0.94	[0.92, 0.96]	0.89	[0.85, 0.92]
BER	0.94	0.90	0.88	[0.83, 0.91]	0.88	[0.84, 0.91]
SC	0.94	0.91	0.92	[0.89, 0.94]	0.80	[0.73, 0.86]
MP	0.96	0.77	0.94	[0.91, 0.95]	0.87	[0.82, 0.91]
Total	0.99	0.96	0.96	[0.94, 0.97]	0.94	[0.92, 0.96]

Note. LA = learning adaptation; SA = social adaptation; BER = behavioral and emotional regulation; SC = self-care; MP = motor performance.

**Table 3 healthcare-13-00148-t003:** Correlation between the COC-15 and the SDQ.

SDQ	LA	SA	BER	SC	MP	Total
Complete sample (*n* = 1085)						
Prosocial	0.67 ***	0.75 ***	0.71 ***	0.63 ***	0.61 ***	0.76 ***
Problematic	−0.63 ***	−0.63 ***	−0.61 ***	−0.51 ***	−0.53 ***	−0.65 ***
Nursery (*n* = 92)						
Prosocial	0.58 ***	0.63 ***	0.41 ***	0.52 ***	0.52 ***	0.65 ***
Problematic	−0.64 ***	−0.50 ***	−0.53 ***	−0.34 ***	−0.37 ***	−0.59 ***
K1 (*n* = 345)						
Prosocial	0.67 ***	0.74 ***	0.69 ***	0.63 ***	0.57 ***	0.74 ***
Problematic	−0.63 ***	−0.63 ***	−0.59 ***	−0.59 ***	−0.55 ***	−0.67 ***
K2 (*n* = 304)						
Prosocial	0.64 ***	0.73 ***	0.68 ***	0.53 ***	0.51 ***	0.71 ***
Problematic	−0.67 ***	−0.63 ***	−0.62 ***	−0.49 ***	−0.51 ***	−0.67 ***
K3 (*n* = 344)						
Prosocial	0.62 ***	0.68 ***	0.67 ***	0.49 ***	0.49 ***	0.70 ***
Problematic	−0.56 ***	−0.61 ***	−0.62 ***	−0.53 ***	−0.50 ***	−0.67 ***

Note. LA = learning adaptation; SA = social adaptation; BER = behavioral and emotional regulation; SC = self-care; MP = motor performance. *** *p* ≤ 0.001. Nursery class is for 2-year-olds; K1 class for 3-year-olds; K2 class for 4-year-olds; and K3 class for 5-year-olds.

**Table 4 healthcare-13-00148-t004:** Differences in the COC-15 scores of children based on service status group.

Service	Complete Sample (*n* = 1085)	Nursery (*n* = 92)	K1 (*n* = 345)	K2 (*n* = 304)	K3 (*n* = 344)
No Service	45.28 (*n* = 720)	28.56 (*n* = 70)	40.59 (*n* = 242)	49.10 (*n* = 203)	52.75 (*n* = 205)
Tier 1	34.89 (*n* = 180)	22.20 (*n* = 20)	30.49 (*n* = 57)	35.98 (*n* = 51)	43.52 (*n* = 52)
Tier 2	36.34 (*n* = 185)	16.00 (*n* = 2)	25.26 (*n* = 46)	35.84 (*n* = 50)	42.94 (*n* = 87)
Tier 1/Tier 2	35.62 (*n* = 365)	21.64 (*n* = 22)	28.16 (*n* = 103)	35.91 (*n* = 101)	43.16 (*n* = 139)
Effect size—no service vs. Tier 1	0.81	0.522	0.80	1.40	1.11
Effect size—no service vs. Tier 2	0.70	1.06	1.22	1.32	1.20
Effect size—Tier 1 vs. Tier 2	0.11	0.48	0.41	0.01	0.06
Effect size—no service vs. Tier1/Tier 2	0.75	0.57	0.98	1.32	1.13

Note: Nursery class is for 2-year-olds; K1 class for 3-year-olds; K2 class for 4-year-olds; and K3 class for 5-year-olds. Tier 1 service refers to classroom support and group-based training; Tier 2 service refers to individualized training and therapies. Effect size is reported as Cohen’s *d*.

**Table 5 healthcare-13-00148-t005:** Differences in the COC-15 scores of children based on grade level.

Grade	LA	SA	BER	SC	MP	Total
Nursery (*n* = 92)	7.77	5.86	4.13	3.48	5.66	26.90
K1 (*n* = 345)	8.30	7.35	6.43	6.91	7.88	36.88
K2 (*n* = 304)	9.16	8.66	8.09	9.17	9.64	44.72
K3 (*n* = 344)	9.81	9.55	8.99	10.36	10.15	48.87
Effect size—nursery vs. K1	0.18	0.45	0.70	1.05	0.87	0.73
Effect size—nursery vs. K2	0.51	0.91	1.31	2.05	1.82	1.50
Effect size—nursery vs. K3	0.85	1.36	1.72	3.17	2.16	2.13
Effect size—K1 vs. K2	0.31	0.41	0.52	0.75	0.75	0.61
Effect size—K1 vs. K3	0.58	0.74	0.84	1.29	1.00	1.00
Effect size—K2 vs. K3	0.27	0.32	0.32	0.53	0.25	0.39

Note. LA = learning adaptation; SA = social adaptation; BER = behavioral and emotional regulation; SC = self-care; MP = motor performance. Effect size was reported as Cohen’s *d*. Nursery class is for 2-year-olds; K1 class for 3-year-olds; K2 class for 4-year-olds; and K3 class for 5-year-olds.

**Table 6 healthcare-13-00148-t006:** Results of the ROC analyses of the COC-15 with service status as the state variable.

	AUC ^1^	Sensitivity	Specificity	LR+ ^2^	LR− ^3^	OR ^4^	YI ^5^	Cutoff
Nursery	0.662	0.727	0.443	1.305	0.616	2.118	0.17	30.5
K1	0.756	0.757	0.616	1.971	0.394	4.997	0.373	38.5
K2	0.819	0.772	0.739	2.958	0.309	9.587	0.511	44.5
K3	0.786	0.741	0.693	2.414	0.374	6.458	0.434	49.5

Note: Nursery class is for 2-year-olds; K1 class for 3-year-olds; K2 class for 4-year-olds; and K3 class for 5-year-olds. ^1^ AUC = Area under the ROC Curve; ^2^ LR+ = Positive Likelihood Ratio; ^3^ LR− = Negative Likelihood Ratio; ^4^ OR = Odds Ratio; ^5^ YI = Youden Index.

## Data Availability

Data are unavailable due to privacy, ethical, and funding restrictions.

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
