# Peer review of "Development and Validation of Child Observation Checklist for Early Screening of Young Children with Special Needs"

_healthcare, 2025, doi:10.3390/healthcare13020148_

Round 1

Reviewer 1 Report (Previous Reviewer 1)

Comments and Suggestions for Authors

Thank you for your corrections.

Author Response

Reviewer 2 Report (Previous Reviewer 2)

Comments and Suggestions for Authors

Please find the comments:

1. Please check whether all references are relevant. For example, reference 32 is irrelevant and the sentence where it appears should be reconsidered/deleted. Citing a paper which was published about 20 years ago is unpertinent.

2. How was test-retest conducted? What are the methodological aspects of the test-retest testing (lines 305-306 should be incorporated in lines 251-252)? What was the period between measurements? How many measurements did you have? How were the test-retest data calculated? How participants were recruited? This is still unclear despite the fact that I indicated this in the previous review of this resubmitted paper.

3. What are the confidence intervals for ICC coefficients? What are p-values? This is also apply to section 3.2.3.

4. The CFA figure looks like the SEM figure. Please present this in a common way, as in the current form this figure can be misleading. 

5. Please amend the use of abbreviations as there are a lot of inconsistencies (e.g., CFI, TLI etc.) in many places.

Comments on the Quality of English Language

The are several shortcomings, e.g., "Table 3. Correlation between the COC-15 and the SDQ.". A lack of "the" etc. I hope the authors could reread this paper and amend such mistakes.

Round 2

Reviewer 2 Report (Previous Reviewer 2)

Comments and Suggestions for Authors

Please find the comments:

1. The CFA figure is still presented incorrectly. See for instance this paper how to present CFA figure: Gómez-García, M.; Matosas-López, L.; Ruiz-Palmero, J. Social Networks Use Patterns among University Youth: The Validity and Reliability of an Updated Measurement Instrument. Sustainability 2020, 12, 3503. https://doi.org/10.3390/su12093503

2. Ethics: Please describe in detail why data are not available. Id data are anonymised what is the reason to limit the access to the data?

3. Zeroes before full stops in numbers should be used as it is according to the journal's requirements (see previously published papers). Sometimes authors use zeroes before full stops in numbers, sometimes not. This should be edited.

4. Table 2: columns 4 and 7 should have subheadings.

Author Response

Responses to Comments of Reviewer on Dec. 31, 2024

  1. The CFA figure is still presented incorrectly. See for instance this paper how to present CFA figure: Gómez-García, M.; Matosas-López, L.; Ruiz-Palmero, J. Social Networks Use Patterns among University Youth: The Validity and Reliability of an Updated Measurement Instrument. Sustainability 2020, 12, 3503. https://doi.org/10.3390/su12093503

Response: Figure 2 is compiled in accordance with the style shown in the above publication.

  1. Ethics: Please describe in detail why data are not available. Id data are anonymised what is the reason to limit the access to the data?

Response: The project was a consultancy project from the Social Welfare Department of the Government of Hong Kong Special Administrative Region. We are bounded that the data belonged to the funder and cannot be shared.

  1. Zeroes before full stops in numbers should be used as it is according to the journal's requirements (see previously published papers). Sometimes authors use zeroes before full stops in numbers, sometimes not. This should be edited.

Response: A zero is added to all concerned numbers accordingly.

  1. Table 2: columns 4 and 7 should have subheadings.

  Response: Table 2 is reformatted and “Cronbach’s Alpha” and “Intraclass Correlation Coefficient” are added to indicate the values of the columns on page 8.

This manuscript is a resubmission of an earlier submission. The following is a list of the peer review reports and author responses from that submission.

Round 1

Reviewer 1 Report

Comments and Suggestions for Authors

The manuscript titled Development and validation of the Child Observation Checklist for early screening of young children with special needs is a significant attempt to contribute to the practice of identifying children with developmental disabilities. The topic aligns well with the content of the journal. My comments are as follows:

The title is clear and accurately describes the manuscript’s content. The abstract is concise, well-structured, and provides adequate information about the text's content.

The introduction starts with an informative section on the importance of early identification and the current situation in the local environment. However, the description of the educational and training center, as well as the program, is not connected to the subsequent work or the needs of the study. The purpose of this information should be clarified. The need for adapting the ASQ-3 is adequately described, but the rationale for highlighting and describing the HKECDS, which is not later linked to the instrument being developed, must be explained. The Introduction should also be supplemented with relevant data from the literature related to the topic.

In line 156, it is stated that an analysis of local and foreign assessment tools was performed. It is necessary to describe the process of analyzing and selecting items from the analyzed instruments to arrive at the initial 76 items for this questionnaire.

In the Introduction, there is a section on data processing, which does not belong in this section and should be moved to the data analysis section.

Confirmatory factor analysis was applied to assess the validity of the COC. Since this is a new questionnaire, it would be expected to perform exploratory factor analysis. The criterion validity was checked by analyzing the results of two groups of respondents (line 172). Still, it would be more appropriate to assess criterion validity through correlation with an objective measure, an accepted measure, or other variables.

The stated goal is the development and validation of a new scale. However, the hypotheses use the scale, which is still being developed and validated, to measure correlations with behavioral problems.

The sample size is adequate in the methodology. The use of the SDQ instrument is unclear. If it is considered the "gold standard" for checking validity, psychometric data on this instrument should be provided, as well as the reason for correlating these two instruments for this purpose.

The results are presented in tables and graphs, and descriptions of the results obtained are given at the level of the application of individual statistical methods (as previously commented on).

The discussion repeats the statistical data obtained.

Reviewer 2 Report

Comments and Suggestions for Authors

Please find the comments how to clarify the paper.

1) 1.4 Calculation section (except hypotheses) should be presented in the methodology section. It is definitely not a part of the introduction. At least, it looks uncommon. Please justify or restructure the narration.

2) Please present the Procedure of the study, with the recruitment process etc. In general, please use all standard section in your paper. It seems that the paper has a very uncommon structure.

3) Please make sure that all statistical consideration are described. For instance, the CFA fit indices were not described. Each analysis should be described, with specific statistical considerations.

4) CFA: factor loadings were not presented. Please reconsider and present the basic CFA output (lines 268-274).

5) How was test-retest reliability conducted. with the recruitment process? What was the form of ICC? ICC(2,1)?

6) Please check the paper for typos (e.g., line 322, with "(MacCallum et al., 1996).", etc.).

7) Please indicate a type of correlations, Pearson or Spearman or others.

8) What test for post hoc comparisons was used?

9) Please calculate the effect size for all the analyses where possible.

10) Appendix A: Please present scoring instructions for the questionnaire.

The paper in many cases seems to be unclear, especially if it comes to the methodology. It should be very extensively clarified.

Reviewer 3 Report

Comments and Suggestions for Authors

A well developed and interesting study.

Subchapter 1.4 should be moved to material and method

Have you taken into account children with malformations that can be included in the education systems?
